

# A combined analysis of geomagnetic data and cosmic ray secondaries in the September 2017 space weather phenomena studies

Roman Sidorov[1], Anatoly Soloviev[1,2], Alexei Gvishiani[1,2], Viktor Getmanov[1], Mioara Mandea[4], Anatoly Petrukhin[3], Igor Yashin[3]

[1]Geophysical Center of the Russian Academy of Sciences (GC RAS), 119296 Moscow, Russian Federation
[2]Schmidt Institute of Physics of the Earth of the Russian Academy of Sciences (IPE RAS), 123242 Moscow, Russian Federation
[3]National Research Nuclear University MEPhI (Moscow Engineering Physics Institute), 115409 Moscow, Russian Federation
[4]Centre National d'Études Spatiales (CNES), 2 place Maurice Quentin 75 039 CEDEX 01, Paris, France

*Correspondence to*: Roman Sidorov (r.sidorov@gcras.ru)

**Abstract.** The September 2017 solar flares and the subsequent geomagnetic storms driven by the coronal mass ejections were recognized as the ones of the most powerful space weather events during the current solar cycle. The occurrence of the most powerful solar flares and magnetic storms during the declining phase of a solar cycle (including the current 24[th] cycle) is a well-known phenomenon. The purpose of this study is to better characterize these events by applying the generalized characteristic function approach for combined analysis of geomagnetic activity indices, total electron content data and secondary cosmic ray data from the muon hodoscope that contained Forbush decreases resulting from solar plasma impacts. The main advantage of this approach is the possibility of identification of low-amplitude specific features in the analyzed data sets, using data from several environmental sources. The data sets for the storm period on September 6–11, 2017, were standardized in a unified way to construct the generalized characteristic function representing the overall dynamics of the data sequence. The new developed technique can help to study various space weather effects and obtain new mutually supportive information on different phases of geomagnetic storm evolution, based on the geomagnetic and other environmental observations in the near-terrestrial space.

## 1 Introduction

The geomagnetic field protects the Earth's atmosphere from the solar wind, which consists of charged particles released by the Sun's magnetic field. The interaction between the Sun's and the Earth's magnetic fields is complex and important to understand the changes in the solar–terrestrial environment on time scales ranging from minutes to glacial cycles. Many solar observations and parameters exist, ranging from very long timescales to short timescales, however, sunspots (e.g. Solanki, 2003) remain the most prominent indicator of both solar magnetism and solar activity. Now the sunspot cycle has been reconstructed over millennia (Usoskin, 2017) and it represents a manifestation of internal processes best observed in solar magnetic phenomena. The sunspot numbers are generally correlated with both solar flares and Coronal Mass Ejections



(CMEs) rate. These two classes of the largest solar eruptive phenomena are the primary causes of space weather events, we are interested here.  They are frequently, if not always, linked to each other, and are the product of instabilities in the sun multi-scale dynamic magnetic structures. The dynamic processes in the Sun and its atmosphere propagate into interplanetary space to the Earth's near-space. The medium of propagation is the solar wind, a plasma that becomes supersonic as it

expands from the solar corona to fill the heliosphere, as first proposed by Parker (1958), and then extensively described and documented by e.g. Balogh et al. (2014).

Radiation and particles emitted by the Sun, with variable delays, interact with the Earth's magnetic field and the atmosphere, and cause electrical currents to flow in regions of the ionosphere and magnetosphere. The solar wind, travelling into the space, complicates more the near-Earth environment, and mainly its magnetic environment. The Earth's magnetic field

comprises contributions from sources inside the Earth (internal contributions, including those from the liquid core and lithosphere) and outside it (external contributions, including those from ionosphere, magnetosphere and their coupling). The external sources induce secondary fields on the Earth. Important to note is that the core magnetic field acts like a shield to the solar wind that the Sun continually emits. Thereafter, to understand the Sun-Earth environment evolution, the variations of the geomagnetic field need also to be understood (e.g. Mandea and Purucker, 2018).

The temporal variations of the geomagnetic field are mainly described from the ground-based magnetometer measurements. Here, we are interested in rapid variations of the field, linked to the Sun-Earth interactions. They are described by the so-called geomagnetic activity indices. Among them, two are widely used to characterize the geomagnetic conditions, Kp and Dst indices. The Kp, a geomagnetic three-hour-range index, is derived from the standardized K index of 13 magnetic observatories and is intended to measure the solar particle radiation by its magnetic effects. The Dst index is derived from

data provided by 4 near-equatorial geomagnetic observatories and is designed to measure the intensity of the globally symmetrical equatorial electrojet (the "ring current"). Dst is then a measure of geomagnetic activity and is used to assess the severity of magnetic storms. For more details, the reader can access the International Service of Geomagnetic Indices (http://isgi.unistra.fr/).

Space weather is defined as the collection of physical processes, beginning at the Sun, and ultimately affecting human

activities on Earth and in space (Natural Resources Canada 2017). The geometry of the geomagnetic and interplanetary magnetic fields, and their evolution in space and time contribute to the complexities of space weather observed at the Earth's surface, and in the near- Earth space. The electrical currents, and the changing energetic particle populations result in geomagnetic variations, aurora, and can profoundly affect a number of technologies. There are well-known examples of disastrous consequences of some of these disturbances. Induced currents caused by magnetic storms can lead to saturation,

overheating and damage of the high-voltage transformers in electrical substations. In this study we precisely investigate the September 2017's geomagnetic storms linked to an important solar activity observed as Martian aurora (Xu et al., 2018), and across the globe on Earth. The September 2017 magnetic storm driven by the CME also was followed by multiple radio blackouts on the daylight side of the Earth (NOAA, 2017); a similar accident occurred after the storm driven by the X9.0-



class solar flare in 2006. Nowadays, thorough and multifactor space weather monitoring is required to prevent the damages from the destructive space weather impact on the technological systems on the Earth (e.g., Gvishiani et al., 2016a).

To take into consideration these geomagnetic storm aspects, the remainder of the paper is organized as follows. Section 2 presents the different datasets used in the current study. In Section 3 some information on the applied method and its

advantages is given, mainly to show the possibilities to apply it for rapid variations as geomagnetic storms are. Section 4 discusses a specific case study, focused on the September 2017 magnetic storm, and finally Section 5 draws some conclusions and maps out future directions.

## 2 Data

In this study, we analyzed the space weather parameters and geomagnetic activity indices along with the time series from the

URAGAN ("Ustanovka RAspoznavainya Grozovykh ANomaliy" – "Thunderstorm Anomaly Recognition Set" in Russian) muon hodoscope data (Barbashina et al., 2008). Muon diagnostics of the near-Earth space is a modern technology applied in various problems of cosmic ray physics, ionospheric and geomagnetic studies. URAGAN is a wide-aperture, large-area multilayered muon hodoscope (Barbashina et al., 2008), which was designed for studying various phenomena in the circumterrestrial space and, in particular, in the Earth's atmosphere (for example, Mikhaylenko et al., 2011; Barbashina et

al., 2017) and magnetosphere (Astapov et al., 2017), that cause variations in the muon flux at the ground level. The hodoscope provides recording of the muon flux intensity from different azimuthal directions (0°–360°) and zenith angles (0°–80°) with a high degree of spatial and angular accuracies (1 cm and 1°, respectively) (Barbashina et al., 2008). It enables registration of each muon along with its track reconstruction, as well as every 1-minute construction of two-dimensional angular matrix displaying muon flux from the observed hemisphere. For the current study, we reassembled the data matrices

from 1-minute to 1-hour by a simple averaging. Variations of the muon flux in the near-Earth space, which are the secondary cosmic ray data, are associated with the physical processes taking place at the time of anomalous space weather events. During the quiet periods, the muon flux intensity at a ground level is stable and relatively high. Before major geomagnetic storms driven by CME, a strong and rapid decrease can be seen in the muon intensity time series. The effect, called the Forbush decrease (Cane, 2000), occurs due to the screening of the cosmic ray flux by the plasma cloud heading towards the

Earth's magnetosphere. The muon data from URAGAN hodoscope has been successfully used for studying the CMEs (Astapov et al., 2017) and corresponding Forbush decreases.

Fig. 1a depicts the muon intensity values extracted from a central cell of each data matrix from September 6, 2017, 00:00 UT to September 11, 2017, 23:00 UT. Fig. 1b displays the muon intensity time series for corner cells of the matrices. The maximal intensity comes from the zenithal direction to the central area of the hodoscope, thus, the muon intensity data from

the central cell is of the most interest.

We also used hourly sampled Total electron content (TEC) data which is available at the IZMIRAN ionosphere weather database portal (http://www.izmiran.ru/ionosphere/weather/storm/index.shtml). In the GNSS signal processing practice, TEC

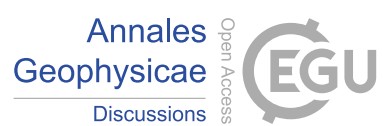

represents the total number of electrons between transmitter and receiver. TEC data is widely used to characterize conditions of the ionosphere and to estimate the ionosphere impact on a radio wave signal (Garner et al., 2008). The more electrons in the path of the signal, the more it is affected. As TEC responds to varying solar electromagnetic and geomagnetic activity (de Haro et al., 2002; Zolotukhina et al., 2017), it can be definitively used in a comprehensive analysis of the space weather

phenomena.

In addition, we studied Dst index time series over the considered period. As seen from Figs. 1c and 1d, the TEC time series appears to be almost a negative reflection of the Dst index, which indicates their similar response to the equatorial ring current.

Also for the demonstration of the onset and evolution of the geomagnetic storm, we used common space weather data from

the OMNIWeb database (http://omniweb.gsfc.nasa.gov/), such as the interplanetary magnetic field (IMF), the solar wind speed, density and dynamic pressure for the case study period of September 6-11, 2017. The IMF and solar wind data are 5-min-averaged. These data are plotted in Fig. 2. Unfortunately, the data are missing for September 9-10 and also there are small gaps of unknown origin in the data time series at the beginning of September 7. The gap in the solar wind data from 3:40 to 5:20 UT is most likely caused by some failure of the particle analyzer.

As the maximal Kp index values during the storm were 8+ according to the GFZ Potsdam data (http://www.gfz-potsdam.de/sektion/erdmagnetfeld/daten-produkte-dienste/kp-index/), the storm was classified as 'severe' (G4 rate) according to the NOAA space weather scale (http://www.swpc.noaa.gov/noaa-scales-explanation). The first local extremum of Kp reached 8− at the end of September 7; after that Kp decreased to its minimal value 0o on September 9 between 18:00 and 21:00. According to the 5-level classification scale based on the Dst index (Loewe and Prölss, 1997), this storm was

classified as 'strong', as the minimum Dst level was –142.

## 3 Method

The proposed technique aims to offer a combined analysis of various space weather related data, including their morphology and time-dependent features. For the needs to better interpret the observations, it is important to identify the particular

features of a physical signal, sometimes difficult to reveal due to their low amplitude. In this case, the generalized characteristic function approach (Troyan and Kiselev, 2010) can be used. This approach is widely implemented in exploration geophysics. For example, it is used for complex interpretation of geoelectric and geomagnetic survey data in the ore deposit search. Here, we applied this new approach in the space weather data analysis.

The generalized characteristic function is dimensionless and thus it enables comparison of data of different origin,

dimensions and scales (ranges of variability) in a single scale of magnitudes. A common formula representing such function for some data sets $f_1, f_2, \ldots, f_n$ over the same time period or the same spatial coordinates is the following Eq. (1):

$$F(f_1, f_2, \ldots, f_n) = \sum_{i=1}^{n} a_i f_{s_i}, \tag{1}$$



where $f_{s_i}$ is the standardized data set for an initial data set $f_i$, and $a_i$ are the weight coefficients depending on the properties of a particular data set, its physical origin and veracity; $i = 1, ..., n$.

Construction of the generalized characteristic function requires determination of typical distribution laws for all data sets under consideration. For example, some physical data can be distributed normally, and some can have the lognormal

distribution law which is defined by mean and standard deviation of the value logarithms. The standardization procedure requires the computation according to Eq. (2):

$$f_s = \frac{f - \bar{f}}{\sigma_f} \qquad (2)$$

In this formula, $\bar{f}$ is the mean $f$ value for a whole data set, and $\sigma_f$ is the standard deviation. In case of a lognormal distribution for $f$, the $\bar{f}$ and $\sigma_f$ values are calculated for the $\ln f$ values.

Several criteria allow establishing the coincidence or inconsistency between estimated distribution law of selected data set and a priori known (reference) distribution law. They are calculated from the corresponding distribution function values and then estimated against a specified threshold. We used the Kolmogorov-Smirnov (KS) test (De Smith, 2015), which enables comparison of the two distributions using their maximum deviation. At first, the distribution of the analyzed data set $X = (x_1, ..., x_N)$ is built. The range of the $X$ variability is divided into $A$ intervals with a width $W_x = (\max(X) -$

$\min(X))/\sqrt{N}$ each. Then, the cumulative distribution function is calculated. For two data sets $X = (x_1, ..., x_N)$ and $Y = (y_1, ..., y_N)$, the KS test value is determined following Eq. (3):

$$D_{XY} = \max_{1 \le i \le A} |c_{Xi} - c_{Yi}|, \qquad (3)$$

where $c_{Xi}$ and $c_{Yi}$ are the cumulated occurrences for the analyzed data sets $X$ and $Y$, respectively. Given a data set of more than 40 values and 5% error probability, Eq. (4) represents the estimate for the critical value $D_C$ of the KS (Troyan and

Kiselev, 2010; Estimation, 2018):

$$D_C = 136/\sqrt{N}. \qquad (4)$$

If $D_{XY} \le D_C$ then distribution laws of two data sets are considered coincident. Therefore, in order to compare calculated value $D$ and the critical value of the KS test $D_C$, it is needed to generate a reference cumulative distribution function using the specified data intervals, mean and standard deviation. As a result, we decided whether the distribution law of the selected

data set and the reference distribution law are coincident or not.

The described approach to the analysis for multiple data sets of different origin is close to widely implemented techniques based on correlation calculations between different data sets. Multiple correlations are more focused on closeness of the patterns in the analyzed data, whereas the generalized characteristic functions approach provides a data combination with a reference to normal (background) levels for each data set. Also one of the main advantages with respect of other analysis

methods is a possibility of identification of low-amplitude patterns related to the specific features of a physical process in all the analyzed data sets, using data from several environmental sources. Another significant advantage is its flexibility: it is possible to adjust the data set contributions using weight coefficients with the standardized time series. As mentioned earlier,



depending on the specialties of some additive components, they can be included into the generalized function with a negative sign, which enables setting several characteristic functions instead of one and further analysis of their mutual behavior.

With the application of the generalized characteristic function approach to such particular space weather parameters as geomagnetic Dst index, TEC and cosmic ray secondary data, we highlight the particular advantages of this technique that can be useful in future space weather studies.

## 4 Case study (the September 2017 magnetic storm)

The September 2017 space weather phenomena driven by two solar flares, provide an opportunity for a study of geomagnetic activity along with the secondary cosmic ray monitoring data. The overall evolution of the September 6–11, 2017 space weather events included several steps. The first solar flare on September 6 occurred at 09:10 GMT. It blasted from a large sunspot on the Sun's surface and was rated X2.2. X-class solar flares are of the most power, according to the solar storm scale of the NOAA Space Weather Prediction Center (NOAA, 2018). The next flare at 12:02 GMT was rated even stronger — X9.3, and it was the biggest X9 flare since an X9.0 event in 2006 (Struminsky and Zimovetz, 2010).

As clearly seen from the IMF time series (Fig. 2), the storm sudden commencement occurred near 00:00 UT on September 6 to 7. At that time, the solar wind speed increased from 400 km/s to about 600 km/s, and there was also an increase in both proton density and flux pressure. The onset of the storm was due to the CME from the second solar flare, which reached the magnetosphere at the end of September 7. This CME produced (1) a higher solar wind speed, which increased over 800 km/s at the beginning of September 8 (Fig. 2b), and (2) a spike-shaped increase of proton density and plasma flux pressure at about 00:00 – 02:00 UT on September 8 (Fig. 2c and 2d).

In this research, we used hourly sampled muon data corrected for the variations of pressure and temperature of the Earth's atmosphere in order to reduce the variability of muon intensity. However, it was still contaminated by variations, and therefore the trend for this time series was built using a piecewise-linear approximation. Black curve in Fig. 1a represents the trend component.

Initial inter-comparison between muon, Dst index and TEC data shows that the beginning of the Forbush decrease is close to the storm sudden commencement (SSC). It is clearly seen in Dst plot as an uplift (Fig. 1c) and in TEC plot as a decrease at the very beginning of September 7. During the period of the storm onset and main phase, the Forbush decrease trend is close to linear. Very slight and insignificant oscillations of the muon flux intensity trend are seen on September 8, during two local extremums of the Dst index (−142 and −122 nT) and TEC (both above 6 TEC units) time series at the beginning and at the end of the day. The muon intensity trend (black curve in Fig. 1a) remains quasi-constant until the 2nd half of September 9 when the recovery phase of the storm was approaching to the end. Therefore, the impact of the CME caused by the second X9.3 solar flare appeared to be not so strong comparing to the impact of the first CME.

TEC increase on September 10, 00:00 UTC, and its consequent abrupt decrease in the morning of September 10 are practically not reflected in Dst values, which steadily increase to zero. At the same time, the muon intensity begins to





increase 2-3 hours before September 10 midnight. It reflects the processes that can definitely have the same origin related to the change of conditions in the ionosphere due to the decay of the geomagnetic disturbance. This issue demands further research in order to study the opportunity of application of muon intensity data to adjust determination of time moments of geomagnetic storm decay.

Fig. 3 shows the results of the distribution law determination for all used data sets. We checked the assumptions of normal and lognormal distributions. As the data lengths are equal, due to their equal sampling, the $D_C$ value for the KS is 11% (eq. (2)). The KS test shows that the muon flux data from the central cell of the URAGAN hodoscope has a distribution very close to lognormal (Fig. 3a) with $D = 2\%$ against $D_C = 11\%$. The KS test for the trend component of the muon flux intensity time series (Fig. 3b) shows less coincidence (14%), however the KS test for a normal distribution of a muon trend shows an
even worse coincidence (more than 20%). Therefore, we accepted the lognormal distribution for the muon intensity trend. The Dst index values appear to be distributed normally with $D = 7.3\%$ (Fig. 3c), and the KS test for the TEC data set suggests its lognormal distribution, although the $D$ value is equal to critical ($D = 10.8\%$) (Fig. 3d). Therefore, the generalized characteristic function should include the original values of Dst index and the natural logarithms for muon intensity and TEC data.

In this research we calculated the means and the standard deviations for the whole observation period. As mentioned before, the Dst index time series has a normal distribution. Despite the standardized values are dimensionless, they still keep the initial time series morphology and therefore can be included in the generalized function as additive components. For the time series having lognormal distribution (muon intensity and TEC data), the mean and the standard deviation are calculated from the value logarithms.

Therefore, we used two generalized functions in order to estimate the correlation between 1) the cosmic ray secondary data and geomagnetic data; and 2) the cosmic ray secondary data and TEC data. Mutual analysis of two generalized functions provides a better identification of possible fragments of interest, i.e. their deviations from each other. In this study, we assumed that the muon flux data, Dst index and TEC data are of the same confidence, and, therefore, the weight coefficients for all the standardized time series are taken 1. However, it is clearly seen that the TEC time series morphology commonly
repeats the Dst time series with a negative sign, which suggests taking weight coefficient −1 for standardized TEC data. The resulting generalized characteristic function is defined following Eq. (4):

$$G_1 = Mu_s + Dst_s,$$
$$G_2 = Mu_s - TEC_s \tag{4}$$

where $Dst_s$, $Mu_s$ and $TEC_s$ are the standardized Dst index, standardized logarithms of muon intensity trend and TEC
values, respectively.

Fig. 1e shows the mutual behavior of $G_1$ and $G_2$ functions. The generalized characteristic functions represent the overall dynamics of muon intensity, Dst index and TEC. Therefore, such an approach can provide features of space weather disturbances derived from geomagnetic, ionosphere and secondary cosmic ray data. Weight coefficients for particular time



series, included into the generalized characteristic functions, can be customized in order to reveal specific patterns of their behavior, such as uplifts, mistiming and others.

## 5 Discussion and Conclusions

As seen from the data plots for the whole observation period, the muon intensity trend remains unchangeable and close to linear during the main phase and recovery phase of the geomagnetic storm. TEC data commonly represents the negative reflection of the Dst index, with a response to the storm sudden commencement and impacts from the coronal mass ejections. Generalized function $G_2$, which characterizes mutual behavior of muon and TEC data, shows a series of slight uplifts closer to the end of the storm. The presence of these uplifts indicates a change in the ionosphere, due to the decay of the geomagnetic disturbance. This particularity however is not represented in the generalized function $G_1$, which characterizes mutual behavior of muon data and Dst index, as seen from the behavior of both functions during September 10–11.

Herein, we constructed two functions because Dst morphology repeats TEC morphology with a negative sign. Setting up two synchronized functions helped us to reveal typical patterns of the time series on the basis of mutual deviations of the generalized characteristic functions from each other.

Finally, let us summarize the main findings of this study. Firstly, we introduced the generalized characteristic functions for integral estimation of space weather disturbances derived from geomagnetic, ionosphere and secondary cosmic ray data. We calculated these functions over the period of September 6-11, 2017 in order to test our approach in the geomagnetic storm conditions. The results show that the proposed approach can be used for identification of one or more patterns representing the same physical process in space weather. Also, as mentioned, the muon flux intensity increase during the decay phase of the storm can be related to the change of conditions in the ionosphere. Fragments of incoherency between the $G_1$ and $G_2$ at different phases of the storm duration indicate the relative contributions from the parameters analyzed and their mutual alternation.

Summarizing all the results, the generalized characteristic functions approach is a combined analysis tool for different space weather data sets that can provide mutually supportive information on the onset, evolution and recovery phase of a geomagnetic storm. It is most likely applicable to estimating separate contributions of geomagnetic, ionosphere and cosmic ray origins during evolution of geomagnetic storm. Moreover, this approach provides an opportunity to apply muon and TEC data as a support for the determination of time moments of geomagnetic storm decay. The creation and update of storm catalogues is often performed using geomagnetic activity indices. Various circumterresrial space monitoring tasks can require even 1-minute temporal resolution, and most of the classical indices provide a resolution of 1 hour and less. Therefore, an improvement of the storm catalogization procedure is needed, using additional environmental data as a support. We consider that the needed efficiency can be achieved by the analysis of 1-minute sampled geomagnetic data over a set of geomagnetic observatories (or some geomagnetic index data of resolution higher than 1 hour, such as 1-min

SYM/ASY indices) in combination with 1-minute muon flux intensity time series. This research describes a primary result. The method should be tested on data registered during quiet geomagnetic activity periods and on other geomagnetic storm events to improve the results and make further conclusions. This will be the purpose of our future studies.

## 6 Author contribution

The main conception of the collaborative work of the GC RAS and MEPhI on the development of a method of early detection of geomagnetic storms based on digital processing of muon hodoscopes observations matrices was formulated by Prof. V. Getmanov and Prof A. Gvishiani (GC RAS) and Prof. A. Petrukhin and Prof. I. Yashin (MEPhI). Prof. A. Petrukhin is a scientific director, ahd Prof. I. Yashin is a deputy head of the Scientific & Educational Centre NEVOD of the MEPhI, which includes the URAGAN muon hodoscope.

The method was developed by R. Sidorov and Dr. A. Soloviev and improved and optimized by Prof. V. Getmanov who added a smoothing procedure for the muon data.

Dr. M. Mandea wrote an introduction and organized the paper in the most successful way and provided some references on recent researches related. Prof. I. Yashin gave valuable advice on some solar activity phenomena.

## 7 Competing interests

The authors declare that they have no conflict of interest.

## 8 Acknowledgements

The results presented in this paper rely on data collected at the INTERMAGNET magnetic observatories. We express our gratitude to the national institutes that support them and the INTERMAGNET community for promoting the high standards of magnetic observatory practice (http://www.intermagnet.org). We also acknowledge URAGAN muon hodoscope data
provided by the National Research Nuclear University MEPhI, and total electron content data provided by IZMIRAN ionosphere weather portal. Facilities of GC RAS Common Use Center "Analytical Center of Geomagnetic Data" were used for conducting the research. The research has been conducted in the framework of the Russian Science Foundation project No. 17-17-01215.

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





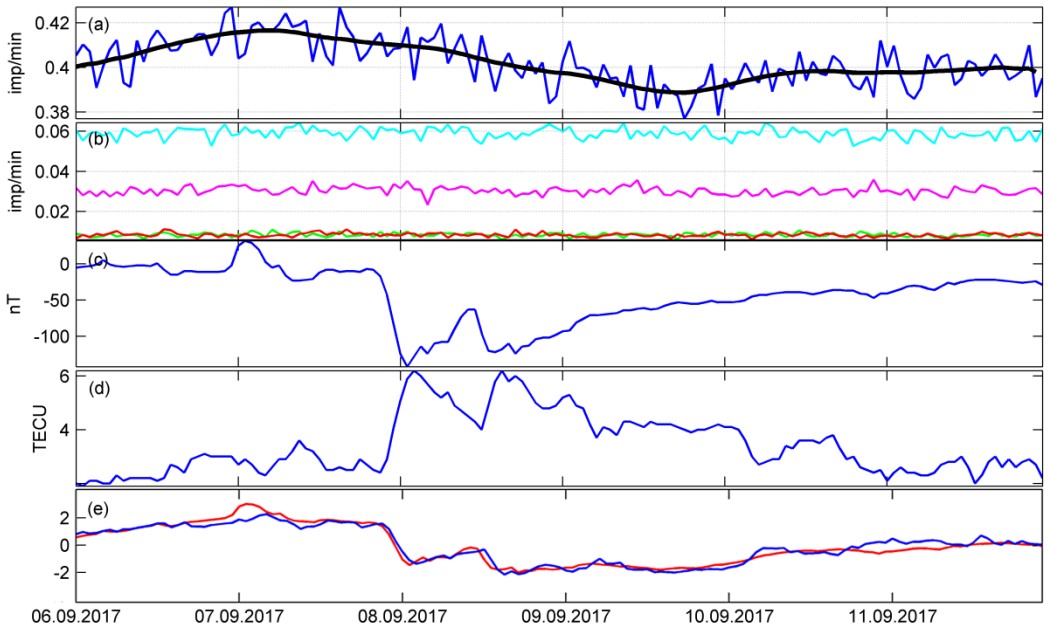

**Figure 1: The inter-comparison between the muon intensity data from the center of the muon hodoscope matrices (a), from the four corners of the hodoscope matrices (b), the Dst index (c), total electron content (d) and the generalized characteristic functions (e) G1 (red) and G2 (blue).**

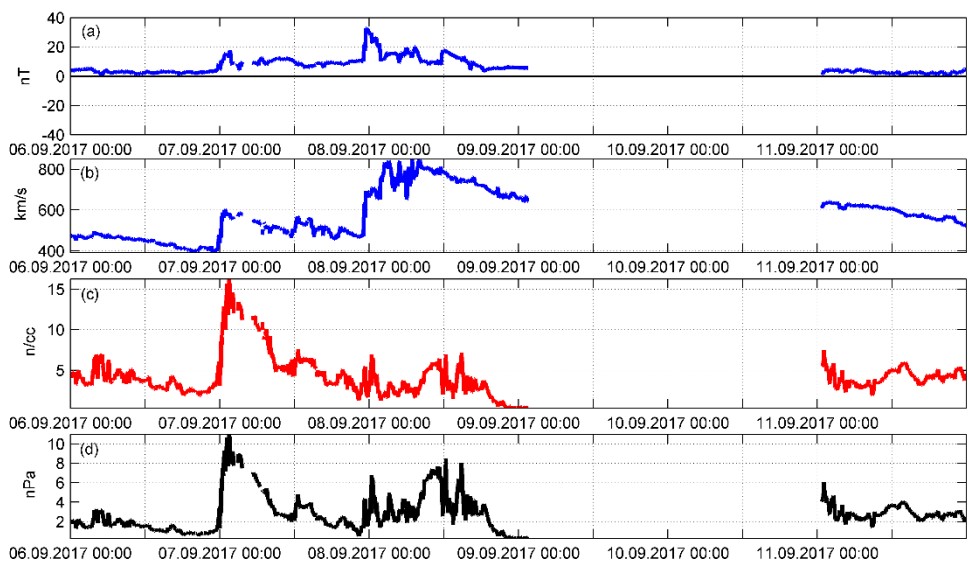

**Figure 2: Solar wind parameters during the solar flares and geomagnetic storm September 6-11, 2017: interplanetary magnetic field (a), solar wind speed (b), proton density (c) and plasma flux pressure (d).**





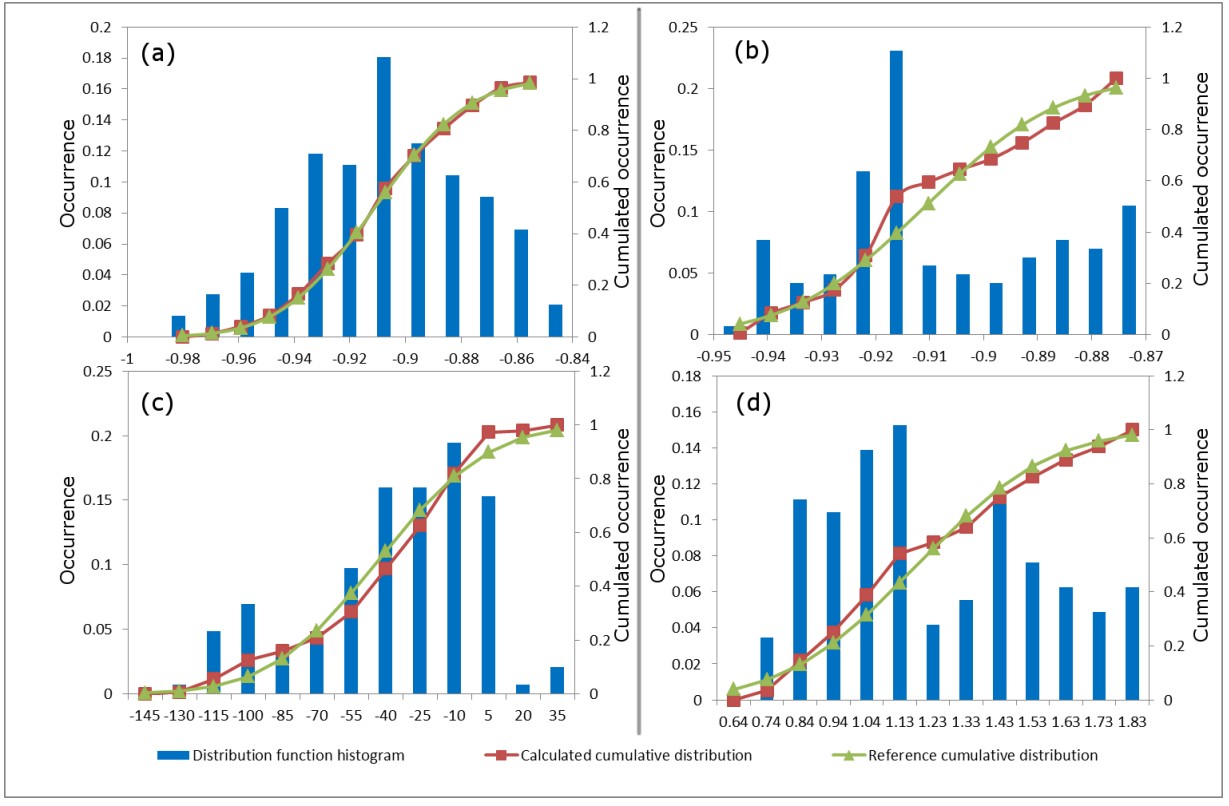

**Figure 3: The results of statistical calculations for determination of the distribution laws for the muon flux (a), muon flux trend component (b), Dst index (c) and TEC (d) data sets. For each plot, the left vertical axis shows the distribution function values, and the right vertical axis shows the cumulative distribution function values.**