# Peer review of "A combined analysis of geomagnetic data and cosmic ray secondaries in the September 2017 space weather phenomena studies"

_Annales Geophysicae, 2018_

## Referee Comment (RC1) · Anonymous Referee #1 · 21 Nov 2018

Main point of concern:

It is not clear (to me), from the explanation of the statistical analysis given in sections 3 and 4, if the authors are analyzing only data for September 2017 or if it possibly includes other data. Note, if it is only for data from September 2017, I then wonder if the authors are using all of the data for this month and whether or not they have removed its autocorrelation.

It is important to recognize that autocorrelated time series (such as space-weather indices covering only a single month) are not statistical data (which are assumed to be independent). In this respect time series analysis and statistical analysis are fundax

mentally different. It would, therefore, be a serious mistake to simply lump all values of a time series into a statistical analysis and fit distribution functions to them, and, even, apply tests of significance (as the authors do with Kolmogorov tests). I refer the authors to the following reference material:

Priestley, M. B., 1981. Spectral Analysis of Time Series, Academic Press, London, UK, Chapter 5.3.2.

Thiebaux, H. J., and F. W. Zwiers, The interpretation and estimation of effective sample size, J. Climate Appl. Meteorol., 23, 800-811, 1984.

von Storch, H., Misuses of statistical analysis in climate research, in Analysis of Climate Variability: Applications and Statistical Techniques, edited by H. von Storch and A. Navarra, pp.11-25, Springer-Verlag, New York NY, 1995.

The issue of autocorrelation needs to be clearly addressed before this manuscript is considered acceptable for publication.

Smaller issue:

The abstract of a paper should be a terse summary of results. It should not be an introduction to the article (we have the "introduction" section for that). I don't easily understand what the results are from reading the abstract. This needs to be entirely redone.

---

## Referee Comment (RC2) · Anonymous Referee #2 · 20 Feb 2019

Page and line numbers refer to file angeo-2018-111.pdf, downloaded 21 January 2019.

I apologize to the authors and to the editor for taking so long to prepare these comments.

General comments:

Summary: I would rate the scientific contribution of the manuscript as "may have potential after additional work and resubmission," because while an interesting way of looking at the data is described, it is far from clear to this reader exactly what we are to learn from this, and how we would tell when to have confidence in any such conclusions.

[Figure]

This work brings a new sensor, the URAGAN muon hodoscope, to the study of space weather phenomena, and it describes a mathematical technique, the "generalized characteristic function," that can be used to compare time series of observed quantities (including the hodoscope data) that have different dimensions and dynamic ranges. There may indeed be potential to learn new things about space weather by bringing these to bear on it. However, I did not find sufficient description in the paper of how to use the generalized characteristic function to draw specific, quantitative conclusions about the correlations of such time series.

Specific comments:

The paper spends a significant amount of space on discussing how to "standardize" each time series, normalizing the data to bring all time series of interest into a state (dimensionless, and with similar dynamic ranges) that allows more direct comparison. Lines 3-25 on page 5 and lines 5-19 of page 7, and the entirety of figure 3, separate the time series of interest here into those with normal distribution and those with log-normal distribution, presenting a quantitative test (equations 3-4) for how well these functional forms represent each time series.

But what are we to do with these standardized time series once we have generated them? Equation 1 defines a "generalized characteristic function" as a linear combination of standardized time series, but all that is said about the weight coefficients is that they "depend[] on the properties of a particular data set, its physical origin and veracity." In lines 31-32 of page 5 it is said to be "possible to adjust the data set contributions [to the GCF] using weight coefficients with the standardized time series," and a few lines later it is mentioned that these coefficients can be negative as well as positive, but how does one choose their values? The two GCFs defined in equation 4 on page 7 (which should be equation 5) and plotted in panel (e) of figure 1 have weight coefficients of plus or minus unity; why? Line 33 of page 7 through line 2 of page 8 says that "weight coefficients for particular time series . . . can be customized in order to reveal specific patterns of their behavior, such as uplifts, mistiming and others," but what such

considerations were taken into account to select the +/- 1 coefficients of this work?

And once we have these GCFs, what do we learn by comparing them? No quantitative correlation tests between GCFs are presented, and I do not see how putting the slowly-varying muon trend time series in each GCF tells us anything that we would not see from a simple comparison of the (standardized) TEC and Dst time series. For example, what does the muon data add to the discussion in lines 5-11 of page 8? In panel (e) of figure 1, the "slight uplifts" in GCF G2 and their absence in G1 give us the same information as is seen in comparing panels (c) and (d) directly. In lines 19-20 of page 8 it is said that "the muon flux intensity increase ... can be related to the change of conditions in the ionosphere"; how do we obtain this relation from the GCFs? In lines 20-21 of page 7 it is stated that the two GCFS are "used in order to estimate the correlation between" muons and Dst and between muons and TEC. Where is this estimate presented in the paper? I do not have access to the Troyan & Kiselev book so that I can go look up the GCF technique; lines 26-28 of page 4 say it is "widely implemented in exploration geophysics," but a space weather audience for the paper will need more detailed and quantitative explanations than are provided.

Finally, if questions of timing are to be addressed, the details of how the muon "trend ... was built using a piecewise-linear approximation" need to be given, in order to ensure that any features in the muon trend data whose timing is compared with features in other data sets are not simply artifacts of the way the approximation was constructed.

Technical corrections:

Generally the paper is clear, though additional proofreading after revision would be helpful. Here are a few minor suggestions I noted in passing.

On line 13 of page 1, "the ones of the most powerful ... events" has a few extra words; perhaps "some of the most powerful ... events."

On line 2 of page 5, "veracity" is not clear – reliability? Accuracy?

[Figure]

On page 7, equation 4 should be numbered equation 5.

Reference #10 still has http://dx.doi.org/ attached to its DOI number.
* * *

---

## Author Comment (AC1) · 15 Mar 2019

Dear Anonymous Referee, Thank you very much for your reviews of our manuscript and for all your valuable comments and remarks regarding its improvement and resubmission. This reply will contain also the list of changes in the future revised version of the manuscript according to your comments.

On your comments: 1) "Main point of concern: It is not clear (to me), from the explanation of the statistical analysis given in sections 3 and 4, if the authors are analyzing only data for September 2017 or if it possibly includes other data. Note, if it is only for data from September 2017, I then wonder if the authors are using all of the data for

this month and whether or not they have removed its autocorrelation. It is important to recognize that autocorrelated time series (such as space-weather indices covering only a single month) are not statistical data (which are assumed to be independent). In this respect time series analysis and statistical analysis are fundamentally different. It would, therefore, be a serious mistake to simply lump all values of a time series into a statistical analysis and fit distribution functions to them, and, even, apply tests of significance (as the authors do with Kolmogorov tests). I refer the authors to the following reference material: Priestley, M. B., 1981. Spectral Analysis of Time Series, Academic Press, London, UK, Chapter 5.3.2. Thiebaux, H. J., and F. W. Zwiers, The interpretation and estimation of effective sample size, J. Climate Appl. Meteorol., 23, 800-811, 1984. von Storch, H., Misuses of statistical analysis in climate research, in Analysis of Climate Variability: Applications and Statistical Techniques, edited by H. von Storch and A. Navarra, pp.11-25, Springer-Verlag, New York NY, 1995. The issue of autocorrelation needs to be clearly addressed before this manuscript is considered acceptable for publication".

- All the data sets refer only to the period September 6-11, 2017 during the geomagnetic storm, no other data for that month were used. We did not remove the autocorrelation from the data time series, as the generalized function approach does not require this procedure before the distribution laws determination. The smoothed muon flux intensity data (as well as the raw data) does not seem to contain significant diurnal variation contribution. For Dst index, the autocorrelation due to the contribution of trend or cyclical components is negligible, as the Sq variation is commonly eliminated during the index calculation, and the low-frequency secular variation does not affect the 5-day period data. The same refers to TEC time series which is the result of multiple ionospheric monitoring data processing. Actually we have not done the full statistical analysis but only built the distributions for the data sets and performed the tests for determination of their distribution laws that would be enough. We thank the anonymous reviewer for a valuable comment on the autocorrelation problem. The data preprocessing including the effective size estimation and the autocorrelation removal

should be a part of our further studies in this field, dealing with the analysis of the geomagnetic variations instead of their indices. The statements on the autocorrelation removal problem were added in the text (p.8, line 9–14).

2) "Smaller issue: The abstract of a paper should be a terse summary of results. It should not be an introduction to the article (we have the "introduction" section for that). I don't easily understand what the results are from reading the abstract. This needs to be entirely redone."

- The abstract was shortened and rewritten.

Thank you again for your comments.

Roman V. Sidorov (on behalf of all co-authors).

―――――――――――――――――――――

---

## Author Comment (AC2) · 15 Mar 2019

Dear Anonymous Referee, Thank you very much for your reviews of our manuscript and for all your valuable comments and remarks regarding its improvement and resubmission.

The replies will also include a list of changes in the texo of the future (revised) version of the manuscript.

On your comments:

1) "This work brings a new sensor, the URAGAN muon hodoscope, to the study of

space weather phenomena, and it describes a mathematical technique, the "generalized characteristic function," that can be used to compare time series of observed quantities (including the hodoscope data) that have different dimensions and dynamic ranges. There may indeed be potential to learn new things about space weather by bringing these to bear on it. However, I did not find sufficient description in the paper of how to use the generalized characteristic function to draw specific, quantitative conclusions about the correlations of such time series".

The approach described in the manuscript is a qualitative evaluation tool. Herein, the principal abilities of this technique are presented, and during the future development it can be possible to define some quantitative measures of contrast resulting from the combination of uplifts or decreases on generalized characteristics. The corresponding statements are added in the manuscript (p.4, line 25, p.8, line 28, p.9, lines 16-18), also the origins of the method are included in the reference list.

Specific comments: 2) "The paper spends a significant amount of space on discussing how to "standardize" each time series, normalizing the data to bring all time series of interest into a state (dimensionless, and with similar dynamic ranges) that allows more direct comparison. Lines 3-25 on page 5 and lines 5-19 of page 7, and the entirety of figure 3, separate the time series of interest here into those with normal distribution and those with lognormal distribution, presenting a quantitative test (equations 3-4) for how well these functional forms represent each time series. But what are we to do with these standardized time series once we have generated them? Equation 1 defines a "generalized characteristic function" as a linear combination of standardized time series, but all that is said about the weight coefficients is that they "depend[] on the properties of a particular data set, its physical origin and veracity." In lines 31-32 of page 5 it is said to be "possible to adjust the data set contributions [to the GCF] using weight coefficients with the standardized time series," and a few lines later it is mentioned that these coefficients can be negative as well as positive, but how does one choose their values? The two GCFs defined in equation 4 on page 7 (which should be

equation 5) and plotted in panel (e) of figure 1 have weight coefficients of plus or minus unity; why? Line 33 of page 7 through line 2 of page 8 says that "weight coefficients for particular time series... can be customized in order to reveal specific patterns of their behavior, such as uplifts, mistiming and others," but what such considerations were taken into account to select the +/- 1 coefficients of this work?"

The weight coefficients were chosen equal to 1 in absolute value as we suppose that all the data sets have the same reliability. We introduce a couple of functions for the analysis of their mutual behavior using the similarity in geomagnetic Dst index and TEC data and thus introducing the -1 coefficient before TEC data, as TEC is similar to Dst with a negative sign. The corresponding clarifications were added (see p.4, line 29-31, p. 7, lines 25-31).

3) "And once we have these GCFs, what do we learn by comparing them? No quantitative correlation tests between GCFs are presented, and I do not see how putting the slowly varying muon trend time series in each GCF tells us anything that we would not see from a simple comparison of the (standardized) TEC and Dst time series. For example, what does the muon data add to the discussion in lines 5-11 of page 8? In panel (e) of figure 1, the "slight uplifts" in GCF G2 and their absence in G1 give us the same information as is seen in comparing panels (c) and (d) directly".

Construction of two functions appears to be more descriptive in this case to reveal possible deviations from this similarity (like the one around the midnight of 07.09.2017 or 11.09.2017). In addition, we added the standardized muon flux intensity data to both time series to include its contribution and make a "link" between them. Muon data included in the GCFs reveals the simultaneous alternation of all three physical data sets. The correlation coefficient between G1 and G2 reaches 0.9553. This has been added in the text (see p.8, lines 4-5).

4) "In lines 19-20 of page 8 it is said that "the muon flux intensity increase... can be related to the change of conditions in the ionosphere"; how do we obtain this relation

from the GCFs?"

Both GCFs repeat that slight muon flux intensity increase is associated with the storm decay during 10–11 September 2017. See the remark on p.8, lines 21-24.

5) "In lines 20-21 of page 7 it is stated that the two GCFS are "used in order to estimate the correlation between" muons and Dst and between muons and TEC. Where is this estimate presented in the paper? I do not have access to the Troyan & Kiselev book so that I can go look up the GCF technique; lines 26-28 of page 4 say it is "widely implemented in exploration geophysics," but a space weather audience for the paper will need more detailed and quantitative explanations than are provided".

Thank you for this remark. We used an unclear statement here; indeed, it is better to say that the GCFs are used in order to "analyze the mutual relation" between muon ans Dst index and between muon data and TEC. The generalized characteristic allows a qualitative interpretation, not a quantitative estimate. Nevertheless, future development of this technique will include some quantitative estimates; as we said in the Discussions and Conclusion section, this manuscript presents a primary result of a single case study. The corresponding statement (page 7, line 25) was rewritten.

6) "Finally, if questions of timing are to be addressed, the details of how the muon "trend... was built using a piecewise-linear approximation" need to be given, in order to ensure that any features in the muon trend data whose timing is compared with features in other data sets are not simply artifacts of the way the approximation was constructed".

The local approximation technique used here is presented in (Getmanov V G, Sidorov R V and Dabagyan R A. A Method of Filtering Signals Using Local Models and Weighted Averaging Functions (2015) Measurement Techniques. 58. 1029–1036). We applied the local approximation technique as one of the simplest to reveal the low-frequency component in the muon data. Compared to low-frequency digital filtering, this local approximation does not introduce possible artifacts caused by phase distor-
tions in the resulting time series. So we suppose that the smoothing result was not affected by possible data artifacts.

A brief description of the local approximation technique is added in the paper (p.6, lines 22-27). A reference link is added.

Technical corrections:

7) "On line 13 of page 1, "the ones of the most powerful... events" has a few extra words; perhaps "some of the most powerful... events."

This statement has been changed and moved from the abstract to Section 4 according to the comments of the 1st Reviewer so that the abstract can represent just a terse summary of results.

8) "On line 2 of page 5, "veracity" is not clear – reliability? Accuracy?" "veracity" was changed to "reliablilty" as we meant the overall quality of the data and their ability to represent the behavior of some physical process correctly.

9) "On page 7, equation 4 should be numbered equation 5". Done

10) "Reference #10 still has http://dx.doi.org/ attached to its DOI number". The link has been removed. Some other typos in the reference list have been also corrected.

Thank you again for your comments. Roman V. Sidorov (also on behalf of all my co-authors)